# Optimization of a Quantitative Anti-Drug Antibodies against Infliximab Assay with the Liquid Chromatography-Tandem Mass Spectrometry: A Method Validation Study and Future Perspectives

**DOI:** 10.3390/pharmaceutics15051477

**Published:** 2023-05-12

**Authors:** Erin H. Smeijsters, Kim C. M. van der Elst, Amy Visch, Camiel Göbel, Floris C. Loeff, Theo Rispens, Alwin D. R. Huitema, Matthijs van Luin, Mohsin El Amrani

**Affiliations:** 1Department of Clinical Pharmacy, Division Laboratories, Pharmacy and Biomedical Genetics, University Medical Center Utrecht, 3584 CX Utrecht, The Netherlands; 2Department of Immunopathology, Sanquin Research and Landsteiner Laboratory, Academic Medical Centre, 1066 CX Amsterdam, The Netherlands; 3Department of Pharmacy & Pharmacology, The Netherlands Cancer Institute, 1066 CX Amsterdam, The Netherlands; 4Department of Pharmacology, Princess Máxima Center for Pediatric Oncology, 3584 CS Utrecht, The Netherlands

**Keywords:** monoclonal antibody, immunogenicity, anti-drug antibody, infliximab, anti-tumor necrosis factor, liquid chromatography-tandem mass spectrometry, radioimmunoassay, therapeutic drug monitoring, inflammatory disease, inflammatory bowel disease

## Abstract

Monoclonal antibodies (mAbs), such as infliximab, are important treatment options for different diseases. Immunogenicity is a major risk, resulting in anti-drug antibodies (ADAs), being associated with adverse events and loss of response, influencing long-term outcomes. The development of ADAs against infliximab is primarily measured by immunoassays like radioimmunoassay (RIA). Although liquid chromatography-tandem mass spectrometry (LC-MS/MS) is increasingly utilized across different fields, this technique is currently not used for ADAs against infliximab measurements. Therefore, we developed the first LC-MS/MS method. Stable isotopically labeled infliximab antigen-binding fragments (SIL IFX F(ab’)_2_) were used to bind and measure ADAs indirectly. Protein A magnetic beads were used to capture IgG, including ADAs, whereafter SIL IFX F(ab’)_2_ was added for labeling. After washing, internal standard addition, elution, denaturation and digestion samples were measured by LC-MS/MS. Internal validation showed good linearity between 0.1 and 16 mg/L (R^2^ > 0.998). Sixty samples were used for cross-validation with RIA, and no significant difference between ADA concentrations was found. The methods had high correlation (R = 0.94, *p* < 0.001) and excellent agreement, intraclass correlation coefficient = 0.912 (95% confidence interval 0.858–0.947, *p* < 0.001). We present the first ADA against the infliximab LC-MS/MS method. The method is amendable for quantifying other ADAs, making it applicable as a template for future ADA methods.

## 1. Introduction

Monoclonal antibodies (mAbs) are widely used in the treatment of different diseases. In chronic diseases, such as inflammatory bowel disease (IBD), rheumatoid arthritis and psoriasis, and in an expanding variety of cancers, mAbs have become an indispensable therapeutic option. For immune-mediated inflammatory diseases, anti-tumor necrosis factor-α (TNF-α) agents, such as infliximab (IFX), are among the first choice once mAbs are initiated.

Recently, an increasing amount of evidence has been published that endorses proactive therapeutic drug monitoring (TDM) of mAbs in order to personalize the treatment and adjust the dosage whenever necessary. Proactive TDM is interpreted as predefined scheduled drug-level measurements, independent of therapeutic efficacy, whereas reactive TDM is performed when therapeutic failure is suspected. Syversen et al. showed that proactive TDM during maintenance treatment with infliximab (IFX) has a significant favorable outcome on sustained disease control when compared to reactive TDM. Proactive TDM during the initiation of IFX treatment, however, did not show a significant favorable outcome [1,2]. In addition to the increased effectiveness of IFX treatment, it has also been shown that TDM is cost-effective and even a cost-saving tool when compared to empirical strategies for IBD management without TDM or with a reactive approach [3].

Immunogenicity of mAb, i.e., an immune response against the mAb which may result in the development of anti-drug antibodies (ADAs), is one of the threats impacting the long-term treatment outcome. ADAs are associated with adverse events, increased drug clearance and loss of response which could eventually result in treatment failure [4]. In 2003, based on a prospective cohort study by Baert et al., the presence of ≥8 μg/mL ADAs against IFX before infusion was found to have a significant impact on infusion-related reactions and a shorter treatment response [5]. A meta-analysis in 2013 by Nanda et al. showed that detectable ADA concentrations against IFX resulted in an increased risk of loss of response to IFX in IBD patients [6].

The majority of ADA assays are using an enzyme-linked immunosorbent assay (ELISA), homogenous mobility shift assay (HMSA) or radioimmunoassay (RIA) as an analytical method [7,8,9]. Liquid chromatography-tandem mass spectrometry (LC-MS/MS) has been used to a lesser extent for this purpose. This analytic technique, however, is gaining increasing interest from pharmaceutical industries and clinical laboratories.

The LC-MS/MS technique has already proven to be a robust analytical technique in measuring mAbs. For quantitative biopharmaceutical analysis, the LC-MS/MS has specific properties which could be utilized for this analysis, such as a high degree of accuracy, selectivity and precision through the incorporation of an internal standard [10,11]. In addition, LC-MS/MS measures the signal intensity from a specific molecule and allows for multiplexing during the run. This multiplexing ability offers the opportunity to analyze different analytes simultaneously, making batch runs of different mAbs possible, thus increasing throughput significantly [11]. An attractive option with the LC-MS/MS, which is specifically utilized in protein analysis, is the possibility for signature peptide analysis. A designated peptide for the selected mAb or ADA can be selected for analysis, making specific and selective TDM possible.

Although TDM of ADAs against mAbs has come a long way since their effect on the therapy was first described by Baert et al. in 2003, there are still unmet needs and calls for further harmonization [5,12,13].

As of today, the ELISA, HMSA and RIA have been compared in the quantification of IFX and ADAs against IFX [8,9]. The comparison of the LC-MS/MS to the existing techniques has not yet been described. Therefore, the aim of this study is to develop and optimize an ADA quantification method for the LC-MS/MS and to compare the results with those from the well-established RIA method which is widely adopted in clinical care.

## 2. Materials and Methods

### 2.1. Chemicals and Reagents

Full-length human anti-idiotypic ADA against IFX (clone AbD17841_hIgG1) was obtained as a 500 mg/L phosphate-buffered saline (PBS) solution from Bio-Rad laboratories (Hercules, CA, USA). Infliximab (Remicade^TM^) was obtained from Janssen Biologics B.V. (Leiden, The Netherlands) as lyophilized powder and was reconstituted in distilled water to a final concentration of 10 µg/µL. Stable isotopically labeled (SIL) standard IFX was obtained from Promise advanced proteomics (Grenoble, France). Kiovig^®^ human immunoglobulin (IVIg) 100 µg/µL was obtained from Baxalta (Lessines, Belgium). Protein A magnetic beads (Magne^®^) were obtained from Promega (Madison, WI, USA). FragIT^®^ kits midispin and microspin, containing IgG-degrading enzyme of Streptococcus pyogenes (IdeS), and CaptureSelect^®^ spin columns were obtained from Genovis (Lund, Sweden). TPCK-Trypsin was supplied by Thermo Scientific as a lyophilized powder and was dissolved in acetic acid (50 mM) to a concentration of 10 µg/µL, aliquoted in Eppendorf LoBind Microcentrifuge tubes and stored at −80 °C. Drug-free human plasma was obtained from Bio-Rad (Berkeley, CA, USA). All other reagents and LC-MS/MS-grade mobile-phase solvents were obtained from Sigma (St. Louis, MO, USA).

### 2.2. Preparation of Standards, Internal Standard and Quality Controls

ADA against infliximab working solution with a concentration of 32 mg/L was prepared by diluting 500 mg/L stock solution in drug-free pooled human plasma. Standards at concentrations of 0.1, 0.25, 0.5, 1.0, 2.0, 4.0, 8.0 and 16.0 mg/L were prepared from the working solution by serial dilution in drug-free pooled human plasma. Internal standard IFX working solution (25 mg/L) was prepared by dilution in PBS (0.1% Tween) and stored at −80 °C. Quality control samples (QCs) at the lower limit of quantification (LLOQ) (0.1 mg/L), QC low (2.5 mg/L), QC med (5 mg/L) and QC high (10 mg/L), were prepared from 500 mg/L stock solution and Bio-Rad drug-free human plasma. Aliquots with QCs were also stored at −80 °C.

### 2.3. Instrumentation and Chromatographic Conditions

Sample purification was performed on a Vibramax 100 plate shaker, Heidolph Instruments (Schwabach, Germany). Sample denaturation and digestion were performed on an Eppendorf™ ThermoMixer C (Hamburg, Germany). All experiments were performed on an Acquity UPLC Class I coupled to a Xevo TQ-XS (Waters, MA, USA). The analytical column was Acquity UPLC BEH, C18, 2.1 × 150 mm, 1.7 μm particle size obtained from Waters Corp. and was maintained at 55 °C. The mobile phases were (A) 0.1% formic acid (FA) in water and (B) 0.1% FA in acetonitrile (ACN). The LC gradients in minutes per percentage of mobile phase B were 0.0 (min)/5 (% B), 1.5/5, 9.5/35, 10/90, 11/90, 11.5/5 and 13.5/5. The flow rate was 0.3 mL/min, and the run time was 13.5 min. The MS was operated in positive mode with a capillary voltage of 2.9 kV, desolvation temperature of 600 °C, desolvation gas flow of 1000 L/h, cone gas flow of 150 L/h and nebulizer pressure of 7 Bar.

Signature peptide selection and chromatographic conditions are provided in Table 1.

### 2.4. F(ab’)_2_ Production

SIL infliximab F(ab’)_2_ and Kiovig^®^ F(ab’)_2_ fragments were produced from 100 µg SIL IFX and 10 mg Kiovig^®^ using the FragIT^®^ microspin and midispin kit, respectively, following the included technical instructions [14]. The kit contains Ides enzyme (FabRICATOR^®^) and CaptureSelect^®^ spin columns. Ides cleaves IgG at a specific site below the hinge region generating F(ab’)_2_ and Fc/2 fragments. Since the Ides enzyme is covalently coupled to agarose beads it stays behind after centrifugation. The generated fragments are then processed with a CaptureSelect^®^ spin column containing anti-Fc llama antibodies coupled to agarose beads, which capture Fc/2 fragments, allowing purified F(ab’)_2_ fragments to be collected. After performing the procedure, a final concentration of SIL IFX F(ab’)_2_ of 0.1 mg/mL and Kiovig^®^ F(ab’)_2_ of 2 mg/mL was obtained.

### 2.5. Sample Preparation

ADAs against IFX in human plasma samples were measured with LC-MS/MS indirectly by measuring the amount of SIL IFX F(ab’)_2_ bound to ADAs to IFX that were captured onto Protein A magnetic beads (Figure 1). Protein A magnetic beads were first reconditioned by pipetting 1 mL homogenized slurry in a Lobind^®^ Eppendorf tube. The tube was placed on a magnetic rack and the storage buffer was removed. Then 800 µL 0.1% tween in PBS was added and removed three times to wash and recondition the magnetic beads.

The magnetic beads were once again resuspended with 0.1% tween in PBS. Then, 50 µL protein A magnetic beads were added to a 500 µL Lobind^®^ 96-well plate, followed by 100 µL 0.1% tween in PBS and a 5 µL sample. The mixture was allowed to bind at room temperature for 1 h at 1000 rpm. After binding of the ADA to the protein A magnetic beads, plasma proteins were removed by pipetting using a magnetic 96-well plate separator (MagnaBot^®^) [15]. Then the magnetic beads were washed 3 times with 250 µL 0.1% tween in PBS shaking each time for 1 min at 1000 rpm. Then 150 µL 0.1% tween in PBS was added onto the magnetic beads, followed by 5 µL 2 µg/µL Kiovig^®^ F(ab’)_2_ to block non-specific interactions caused by anti-hinge antibodies [16]. The samples were allowed to bind for 30 min at 1000 rpm. Then, 2.5 µL 0.10 mg/mL SIL IFX F(ab’)_2_ fragments were added, and the samples were incubated for 2 h at room temperature at 1000 rpm. Excess SIL IFX F(ab’)_2_ fragments were washed away 3 times with 250 µL 0.1% tween PBS, shaking each time for 1 min at 1000 rpm. Then 100 µL 0.5% OG in 0.1% FA was added followed by 5 µL internal standard IFX 25 mg/L to elute the ADA bound SIL IFX F(ab’)_2_ fragments.

Samples were mixed for 5 min at 1200 rpm and transferred to a clean 500 µL Lobind^®^ 96-well plate using MagnaBot^®^, leaving behind the magnetic beads. Then, the extracts were neutralized with 10 µL TRIS 1 M and heat-denatured on a ThermoMixer C block heater set at 80 °C for 30 min. Samples were centrifuged at 4000× *g* for 5 min and 10 µL trypsin (1 mg/mL) was added and gently mixed. Then, the plate was placed in the ThermoMixer C block heater set at 37 °C for overnight digestion. Finally, trypsin activity was stopped through the addition of 25 µL 10% FA in ACN and 25 µL was injected and analyzed on LC-MS/MS.

### 2.6. Magnetic Beads

Experiments were performed to determine the optimum volume of magnetic beads (20% slurry) required to bind 5 µL working standard (32 mg/L). Four different volumes of magnetic beads (5, 10, 20 and 50 µL) were examined on an ADA against IFX-spiked (32 mg/L) blank human plasma sample. Each volume was tested in triplicates for increased certainty. The experiment was performed according to the sample preparation procedure described above with 1 h binding time between protein A and the immunoglobulins (IgGs) and ADAs against IFX in plasma, Figure 1.

### 2.7. Aspecific Interaction

Different amounts of Kiovig^®^ F(ab’)_2_ were tested to determine the minimum required amount needed to block aspecific interactions caused by anti-hinge antibodies in plasma. First, 10 different blank human plasma samples were screened running the method described above without the inclusion of Kiovig^®^ F(ab’)_2_ to identify the blank samples with high background signal. Thereafter, the blank sample with the highest background signal was prepared according to the above-described method with variable volumes (2.5, 5 and 10 µL) of Kiovig^®^ F(ab’)_2_ 0.5 mg/mL. Each volume was tested in duplicate.

### 2.8. Validation

The method was validated according to the EMA guideline for bioanalytical method validation and immunogenicity assessment of therapeutic proteins.

The following method performance indicators were evaluated for linearity, LLOQ, selectivity, carry-over, matrix effect, within-run and between-run accuracy and precision.

Stability was tested for QC low and QC high, each for five replicates per day for three consecutive days. Three conditions tested were bench-top stability at room temperature (20 °C), in the fridge (5 °C) and three freeze/thaw cycles (−80 °C to room temperature) [17,18].

Cut-point limit of the assay was determined by analyzing negative human plasma samples for three consecutive days, in coherence with the FDA guideline for immunogenicity testing of therapeutic protein products [19].

### 2.9. RIA Method for Binding Antibodies

The RIA reference method to measure ADA to IFX in plasma was developed by Sanquin (Amsterdam, The Netherlands) [20].

### 2.10. Patient Samples and Statistical Analysis

The remnant plasma of patients treated with IFX at the University Medical Center Utrecht (UMCU, Utrecht, The Netherlands) for rheumatoid arthritis and IBD was used for the method validation. Sixty-seven samples with IFX levels < 1 mg/L were chosen for ADA against IFX analysis. Aliquots were sent for RIA analysis at Sanquin (Amsterdam, The Netherlands), and the remainder was stored at −80 °C before LC-MS/MS analysis. The performance of the RIA method has been described previously [9,20].

The use of remnant material drawn as part of the treatment protocol and with the patient’s informed consent was in accordance with the University Medical Center Utrecht policy and ethical standards.

Wilcoxon signed-rank test was used to compare the ADA concentrations found between tests. The agreement between the methods was determined by calculating the intraclass correlation coefficient (ICC) using the two-way mixed single measures test for absolute agreement. An ICC of 1 means absolute agreement. The correlation between the results was quantified by calculating Spearman’s rho, where an R^2^ of 1 means an absolute correlation between the RIA and the LC-MS/MS method.

Bland–Altman plot was used for visualization of the method comparison, in which the *y*-axis illustrates the difference between the methods and the *x*-axis the average of the measures of ADAs by both methods.

Statistical analyses and visualization were performed using R version 4.1.3.

## 3. Results

### 3.1. Internal Validation Results

Linearity of the LC-MS/MS ADA against the IFX method, using standards ranging from 0.1 to 16 mg/L, was good (R^2^ > 0.998) using weighted (1/x) simple linear regression. As a quantifier for the LC-MS/MS analysis, the signature peptide “YASESMSGIPSR” was selected. This is a unique amino acid sequence specific for IFX, located in the variable light chain. An LLOQ at 0.1 mg/L was achieved with an overall relative standard deviation (RSD) of 14.6%, providing a signal to noise (S/N) of 5.7 at a concentration of 0.1 mg/L, Figure 2.

The cut-point limit of the 50 negative human plasma samples was tested on three different days, resulting in an average cut-point of 0.050 mg/L, Figure 3. This is well within the FDA requirements of 0.1 mg/L [17,18,19].

Freeze/thaw, benchtop (20 °C) and fridge (4 °C) stability, at QC Low and QC High, showed no decrease in concentration during three different days. The average RSD for the matrix effect for the 13 spiked samples were 9.6% and 5.8% at 0.5 mg/L and 10 mg/L, respectively, which is in accordance with the acceptance criteria in the guidelines of <15%. Samples high in lipids, erythrocytes and bilirubin did not show any interference with the measurements. Accuracy and precision were found to be in agreement with the EMA guidelines, see Table 2.

### 3.2. Cross-Validation

A total of 67 remanent plasma samples were used for the cross-validation of the RIA with the LC-MS/MS method. Seven plasma samples were above the upper limit of quantification (ULOQ) of 880 arbitrary units (AU)/mL which equates to 8.8 mg/L for the RIA assay. This conversion between AU/mL and mg/L has previously been described in a cross-validation study between the RIA and ELISA [9]. To further investigate the results of these samples, they were diluted and re-measured. These results of the ULOQ samples correlated poorly and were therefore not included in the data comparison.

Differences between the ADA against IFX concentrations were compared using the Wilcoxon signed-rank test. There was no significant difference between the ADA concentrations when comparing the 60 samples measured by both the LC-MS/MS and the RIA method.

A high correlation was found between the methods (R = 0.94, *p* < 0.001). Moreover, the methods showed excellent agreement by calculating the intraclass correlation coefficient = 0.912 (95% confidence interval [CI] of 0.858 to 0.947, *p* < 0.001).

The Bland–Altman plot shows that a higher ADA against IFX concentration slightly increases the difference between the methods, Figure 4A. This is also shown by the 95% CI of the regression line for the correlation plot, Figure 4B.

### 3.3. Magnetic Beads

The sample preparation procedure used was based on immuno-affinity interaction. The first interaction in the procedure was between the protein A magnetic beads and IgG in the human plasma sample. In order to obtain maximum recovery, it was necessary to optimize the binding condition between these two components. We found that with increasing protein A magnetic beads volume, an increase in signal intensity was obtained. The signal intensity at 20 µL bead volume was 70%, a further increase of 20% was realized when 50 µL bead volume was utilized. This means that the bead volume was increased by 150% from 20 µL to 50 µL, but the signal intensity only increased by 20%. Because of the modest increase in signal intensity, 50 µL magnetic bead volume was chosen as optimum. This is in accordance with the manufacturer, which states that 50 µL bead slurry has a capacity of 250 µg IgG. Since 5 µL human plasma contains on average 50 µg IgG, 50 µL bead slurry would provide approximately five times more capacity than is required.

### 3.4. Aspecific Interaction

Various antibodies, including rheumatoid factors and anti-hinge antibodies, are known to cause aspecific interactions during immunoaffinity sample preparation [16]. Anti-hinge antibodies are IgGs in the plasma that bind F(ab’)_2_ through their hinge region (site of Ides cleavage). This interaction leads to a high background signal in some ADA-negative patients and even in healthy volunteers it is known to cause false positives. To minimize the effect of aspecific interaction caused by anti-hinge antibodies, Kiovig^®^ F(ab’)_2_ was added prior to the addition of SIL IFX F(ab’)_2_. Kiovig^®^ F(ab’)_2_ would be able to saturate the binding sites of these anti-hinge antibodies, and since it does not contain the signature peptides that are targeted with the LC-MS/MS, it does not interfere with the measurements. An increasing amount of Kiovig^®^ F(ab’)_2_ was found to decrease the observed signal. At 5 µg Kiovig^®^ F(ab’)_2,_ the signal obtained from the blank human plasma sample was 40% of the LLOQ (0.1 mg/L) signal. This is a 15-fold reduction in signal intensity as compared to the sample where no Kiovig^®^ F(ab’)_2_ was added. Previous experiments with intact Kiovig^®^ IgG using concentrations from 25 to 150 µg did not result in the suppression of these aspecific interactions. Since the blank sample only provides a false ADA concentration of around 1 mg/L, it was surprising to note that a significantly higher amount, 5 µg of Kiovig^®^ F(ab’)_2_, was required, as compared to the amount of SIL IFX F(ab’)_2_, 0.25 µg used in the assay, to suppress all aspecific interactions. This could be because the interactions that occur between anti-hinge antibodies and the hinge region of F(ab’)_2_ have a high dissociation constant and become readily available. Therefore, to allow for a greater margin of safety, the final method utilized 10 µg Kiovig^®^ F(ab’)_2_ per 5 µL of human plasma sample.

## 4. Discussion

In this study, we present the first LC-MS/MS technique for analyzing ADAs against IFX and optimization of the method. Until now the main methods that have been used for analyzing ADAs are the RIA, ELISA and HMSA methods. Previous studies have extensively used and compared these methods and they are standardized in clinical practice [8,9,21]. The LC-MS/MS method was compared to the RIA method used by Sanquin Research in Amsterdam. In total, 60 samples were tested using both methods and showed good correlation and excellent agreement. The LC-MS/MS method has not been cross-validated with other methods, such as ELISA. However, the RIA method has previously been compared and showed a good correlation and agreement with other assays currently in use for ADA against infliximab quantification. Therefore, we expect that a good correlation and agreement are likely between these assays and the LC-MS/MS method [8,9,21].

Three samples were outliers and were reanalyzed by both methods, as seen in the Bland–Altman plot, Figure 4A. After reanalysis, the same difference between the methods was present. Possible reasons for these differences were anti-hinge antibodies which can be present in some patients and are known to cause aspecific interactions [16]. The RIA method uses polyclonal IgG (IVIG) F(ab’)_2_ in the sample preparation to block the interfering anti-hinge antibodies. We therefore repeated the LC-MS/MS analysis of these four samples by including 50 µg IVIG F(ab’)_2_ per sample and allowed for 15 min incubation time prior to the addition of IFX F(ab’)_2_. However, the differences in the results remained. The addition of ADM F(ab’)_2_ instead of IVIG F(ab’)_2_ also did not resolve the problem. A possible reason for these differences could be the radioactive isotope Iodine that is added to IFX F(ab’)_2_ to allow for detection in the RIA assay. The iodine reacts with tyrosine (Y) or histidine (H) side chains and results in mono-iodination of the imidazole groups or mono and di-iodination of the tyrosine phenolic group. These iodine atoms have a similar mass to the average mass of an amino acid and are known to interfere with target binding [22]. Since IFX F(ab’)_2_ contains 12 Y and 4 H in the variable heavy and light chains, it is likely that the presence of this element might interfere during ADA-F(ab’)_2_ binding, thus leading to lower results for some patients, depending on the target binding site of the ADA. This is one possible explanation. Of importance, both assays did find a clear positive ADA result, which would lead to the same strategy for further treatment options in clinical practice. There have been no discrepancies between results being detectable on one assay and undetectable on the other, making both assays reliable to measure ADAs against IFX and resulting in the same clinical decision making.

ADAs against other mAbs are not expected to cause aspecific interactions and interfere with the current method. ADAs against IFX bind specifically to IFX, since they bind to a specific amino acid sequence only present in IFX. ADAs against other mAbs are therefore not expected to bind to SIL IFX F(ab’)_2_.

There are still some improvements to be made to expand the knowledge and influence of ADAs against mAbs. More understanding regarding the types of ADAs, the principles and influence of early development of ADAs during therapy and ways to prolong mAb treatment by better ADA assays may greatly benefit the long-term disease management of these patients.

One of the questions remaining is the influence of different types of ADAs. ADAs can be differentiated into two types: neutralizing antibodies (NAbs) and non-neutralizing antibodies (non-NAbs). NAbs bind to the active site of the mAb, thus blocking its disease-modulating activity. Non-NAbs do not bind to the active site of the mAb, so this does not nullify its pharmacological properties, but the formed immune complexes could enhance the clearance of the formed complex via endocytosis [23]. Van Schie et al. have previously shown that the vast majority, >90%, of ADAs in ADA-positive patients are neutralizing. Other studies have found discrepancies between NAb and non-NAb concentrations, and these discrepancies can be largely explained due to the fact that assays are not sensitive enough to reliably differentiate the ADAs [24]. To obtain a better understanding of the types of ADAs and their development during the different phases of mAb treatment, i.e., during mAb initiation, stable disease control and eventually loss of response due to ADAs, reliable assays with high selectivity and specificity are a necessity.

Our research group has previously published an LC-MS/MS method to quantify NAb against IFX [11]. The results of the NAb method however have not yet been compared and validated to the current method for total ADAs. Neither has it been clinically evaluated whether positive NAb titers are more indicative for the loss of response to IFX compared to total ADAs. Further research and studies need to be conducted to compare these techniques which have now been developed for the LC-MS/MS and clarify the influence of NAb versus total ADAs. Seeing that it is not entirely clear whether the loss of response to mAbs can be assigned to the undifferentiated group of ADAs as a whole or rather specifically to the NAb. Some studies suggest that the development of NAbs might be more suggestive for the loss of response. As mentioned before, these conclusions can be largely assigned to a problem with the sensitivity of these assays. The LC-MS/MS method however does offer an opportunity to reliably differentiate between NAb and total ADA concentrations. About 60% of the patients who are treated with IFX develop ADAs without having a direct loss of response to the therapy, which means that the possibility to differentiate the type of ADAs could be indicative for the treatment outcome during the different phases of mAb treatment [25,26].

One of the enigmas remaining is the prevention of, or counteracting, the development of ADAs against mAb, prolonging therapy continuation. This goes hand in hand with the temporal evolution of these ADAs, which requires a better understanding of the underlying mechanism of the pathogenesis of ADAs.

One thing to take notice of is the difference between transient and persistent ADAs. Transient ADA expression means that the ADAs will disappear over time and might not contribute to the loss of response to the treatment. Persistent ADAs remain active and eventually lead to low mAb concentrations and loss of response. These transient ADAs make interpretation of early detected ADAs difficult. Van de Casteele et al. showed that ADAs were transient in 28% of the patients with detectable ADAs. Kim et al. measured these transient ADAs in 30% of pediatric patients with IBD in which initial ADAs were found. Although both groups found ADAs during treatment, they did not show worsening of the treatment due to the transient nature of these antibodies, a phenomenon also shown by Steenholdt et al. [27,28,29]. Ungar et al. conducted a prospective observational study and showed that of the patients developing persistent ADAs against IFX, 90% do so within the first year after treatment initiation, while transient ADAs were detected during the entirety of the IFX therapy, having less implication for the loss of response to IFX or adverse events [30]. The above findings suggest that early and reliable detection of ADAs might present an opportunity to counteract this, for example, with the addition of immunosuppressive co-medication. Baert et al. showed that the combination of immunosuppressive co-medication had a protective effect against the formation of ADAs. Initiating immunosuppressive co-medication therefore might be applicable to suppress ADA formation if detected at an early stage [5].

Therefore, a reliable method of detecting ADAs is necessary. Traditional methods for the detection of ADA possess problems because the remaining mAb concentration interferes with the assay, leading to false negative or lower ADA results. Drug-tolerant assays are designed to be specific for the detection of ADAs against mAb, while tolerating the presence of mAb, thereby overcoming the interference caused by mAb and provide more accurate results.

Both methods in this study are drug-sensitive, meaning that detectable IFX drug levels would interfere with the detection of ADA. Previous studies have shown different techniques to overcome this. An example is using an acid dissociation step before the analysis, but this does require additional steps during sample preparation, making it less ideal for the workflow and throughput. Although an acid-dissociation step introduces additional preparation and work, ADA-positive patients were detected earlier using this drug-tolerant technique. On average, drug-tolerant assays detected ADAs against IFX 20 weeks earlier compared to drug-sensitive assays [31,32].

Apart from adverse events and loss of response to IFX due to ADAs, low drug concentration or prior ADAs are related to an increased risk of developing ADAs to subsequent anti-TNF-α agents [33]. Frederiksen et al. showed that patients who switched from IFX to ADM, because of loss of response due to ADAs against IFX, were significantly more prone to develop ADAs against ADM, compared to patients without previous ADAs [33,34]. This phenomenon has also been shown for subsequent mAbs from another class, where an increased risk for ADAs against vedolizumab after prior ADAs to IFX and ADM was found. Switching could eventually lead to a cascade of higher chances of ADA formation and treatment failure [35].

Ligand-binding assays currently remain the most utilized platforms for ADA detection. The addition of the LC-MS/MS technique we present here might benefit the development of future ADA assays to better understand some of the above-mentioned remaining uncertainties.

The LC-MS/MS method compared with other analytic platforms shows a good performance. The LLOQ of the LC-MS/MS method is 0.1 mg/L, this is comparable to the RIA method, LLOQ of 0.12 mg/L. Other methods for ADA against IFX are using mainly ELI-SA, both commercially and in-house developed assays. LLOQs for ELISA methods range from 0.01 to 1.2 mg/L [9,21]. Although the LC-MS/MS has a slightly higher LLOQ compared to some of the ELISA assays, ADA concentrations < 0.2 mg/L are unlikely/not expected to have a clinical impact on the IFX treatment. The current LC-MS/MS method is therefore sufficiently sensitive [36].

The LC-MS/MS platform has unique properties which could be a valuable method for immunogenicity assessments. The LC-MS/MS is able to isotype ADAs, which offers a high specificity while having lower drug interference and false positives [37].

Our research group has already shown the multiplex capabilities of the LC-MS/MS and presented a workflow [10]. Others have shown the multiplex method to quantify different ADA isotypes. In the future, quantification of multiple ADAs against different mAbs in the same assay might enhance efficiency and reduce workload [38].

## 5. Conclusions

In this study, we present the first LC-MS/MS method to quantify ADAs against IFX in human plasma. The method was first optimized to determine the required amount of protein A magnetic beads needed for optimum ADA binding and recovery. Thereafter, known interference from anti-hinge antibodies, which has been shown to lead to aspecific interactions and false positives, was shielded by the addition of an optimized amount of Kiovig^®^ F(ab’)_2_ as shown to be effective in previous ADA methods by the RIA. The method was validated according to EMA and FDA guidelines in regards to sensitivity and selectivity on assessment of therapeutic proteins and guidelines for bioanalytical method validation. The results show an average cut-point of 0.05 mg/L. Furthermore, an LLOQ of 0.1 mg/L was obtained making it sufficiently low for clinical practice. Cross-validation with the RIA assay showed a high correlation and excellent agreement. There is no significant difference between the ADAs against IFX concentrations, although there were three samples that were clear outliers. Here, LC-MS/MS provided a slightly higher concentration as compared to RIA. However, in both assays, these samples were clearly positive for ADAs and the resulting patient treatment strategy based on the two different results would have been the same.

Although we have obtained a broader understanding and knowledge in regards to immunogenicity of mAbs, utilization of ADA measurements in clinical care remains a challenge. This is partly due to the wide variety of assays which are used to detect ADAs with variable selectivity, specificity, possibility for drug-tolerant measurements and differentiating of ADA types. The addition of the LC-MS/MS method we present in this study is an important step to better understand immunogenicity assessments and to incorporate ADA measurements in future clinical care. The LC-MS/MS offers high selectivity and specificity, differentiating between ADAs and, in the future, drug-tolerant measurements. The method which is presented here will also be able to aid and act as a template for future LC-MS/MS methods to detect mAbs and ADAs. The techniques used for ADAs against IFX measurement are amendable for other ADAs which broaden the possibilities for expanding personalized immunogenicity testing. Improving efficiency for laboratories by increasing throughput significantly by batch runs of ADAs and mAbs and measuring multiple analytes simultaneously due to the multiplexing capabilities.

## Figures and Tables

**Figure 1 pharmaceutics-15-01477-f001:**
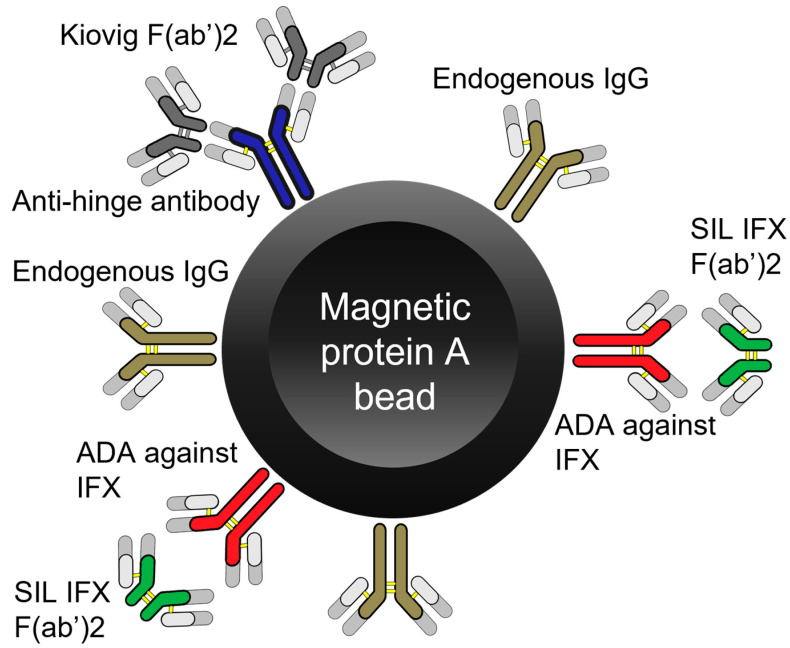
Principle of the test for the indirect measurement of anti-drug antibodies. First, all IgGs including anti-drug antibodies (ADAs) against infliximab (IFX) in serum were bound to protein A magnetic beads. Then, anti-hinge antibodies were blocked with Kiovig^®^ F(ab’)_2_. Then, excess stable isotopically labeled (SIL) IFX F(ab’)_2_ fragments were introduced and allowed to bind to ADAs. Finally, unbound SIL IFX F(ab’)_2_ was washed away and after the inclusion of IFX internal standard, bound SIL IFX F(ab’)_2_ was eluted, and the mixture was denatured, digested and measured on LC-MS/MS.

**Figure 2 pharmaceutics-15-01477-f002:**
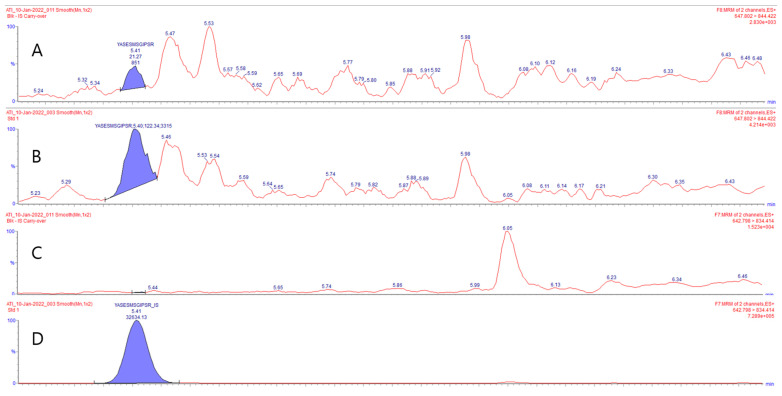
Blank SRM chromatogram of stable isotopically labeled (SIL) YASESMSGIPSR signature peptide (**A**). Standard 1 SRM chromatogram of SIL YASESMSGIPSR signature peptide at LLOQ level (0.1 µg/mL) (**B**). Blank SRM chromatogram of YASESMSGIPSR internal standard (**C**). Standard 1 SRM chromatogram of YASESMSGIPSR internal standard (**D**).

**Figure 3 pharmaceutics-15-01477-f003:**
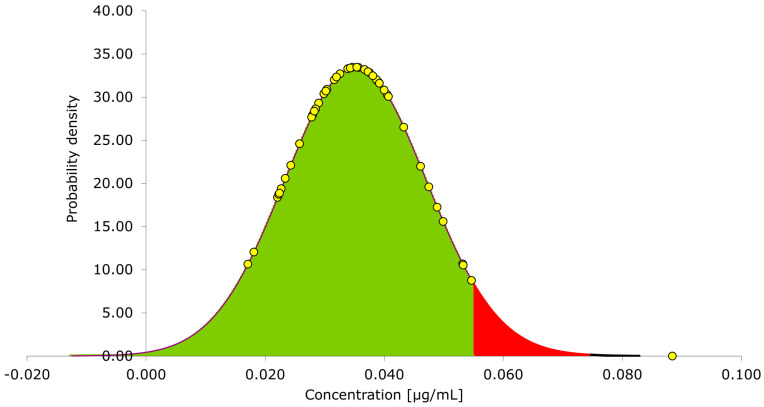
Normal distribution of the anti-drug antibody against infliximab concentration measured in 50 negative human plasma samples as part of the cut-point validation. 95% confidence interval (CI) highlighted in green and 5% chance of false positive (x > µ + 1.65 × standard deviation) highlighted in red.

**Figure 4 pharmaceutics-15-01477-f004:**
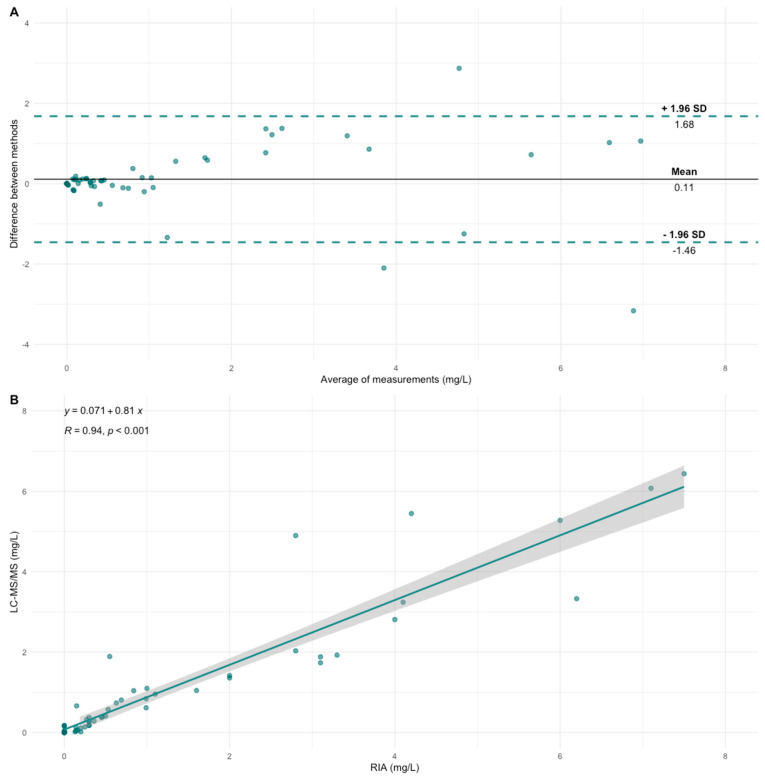
Comparison of anti-drug antibodies (ADAs) against infliximab concentrations measured by the radioimmunoassay (RIA) and the liquid chromatography-tandem mass spectrometry (LC-MS/MS). (**A**) Bland–Altman plot comparing the average ADA concentration in mg/L on the *x*-axis and the difference between the two methods on the *y*-axis. The dotted lines depicted the 95% confidence interval. (**B**) The correlation between the two methods is shown in a scatterplot. The line depicted a regression line with a 95% confidence interval.

**Table 1 pharmaceutics-15-01477-t001:** Xevo TQ-XS SRM transitions and settings for the signature tryptic peptides stable isotopically labeled (SIL) infliximab (IFX) and IFX internal standard (IS).

Peptide Sequence	Used as	Precursor (*m*/*z*)	Product (*m/z*)	Product Ion	Charge	CE ^a^ (V)	CV ^b^ (V)
YASESMSGIPSR[13C_6_,15N_4_]	Quantifier	647.80	844.42	y_8_	1+	25	35
YASESMSGIPSR	IS	642.80	834.41	y_8_	1+	25	35
ASQFVGSSIHWYQQR[13C_6_,15N_4_]	Qualifier	601.96	759.38	y_12_	2+	17	35
ASQFVGSSIHWYQQR	IS	598.63	754.38	y_12_	2+	17	35
SINSATHYAESVK[13C_6_,15N_2_]	Qualifier	472.24	607.80	y_11_	2+	13	35
SINSATHYAESVK	IS	469.57	603.79	y_11_	2+	13	35

^a^ CE, collision energy; ^b^ CV, cone voltage.

**Table 2 pharmaceutics-15-01477-t002:** Accuracy and precision validation data for anti-drug antibodies (ADAs) against infliximab with liquid chromatography-tandem mass spectrometry (LC-MS/MS). Quality controls (QCs) were measured at lower limit of quantification (LLOQ) (0.1 µg/mL), low (2.5 µg/mL), medium (5 µg/mL) and high (10 µg/mL) levels. Within-run data were based on 5 replicates and between-run data on 3 different days.

Precision (% CV)				Accuracy (% bias)
QC	Within-Run	Between-Run	Overall	Overall
LLOQ	12.3	7.8	14.6	14.8
Low	7.4	2.1	7.7	−0.4
Med	6.3	0	6.3	7.4
High	5.9	3.1	6.7	7.8

## Data Availability

Data are available on reasonable request from the corresponding author.

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
