# Peer review of "Optimization of a Quantitative Anti-Drug Antibodies against Infliximab Assay with the Liquid Chromatography-Tandem Mass Spectrometry: A Method Validation Study and Future Perspectives"

_pharmaceutics, 2023, doi:10.3390/pharmaceutics15051477_

Round 1

Reviewer 1 Report

The topic is interesting but there are many articles related to infliximab measurement (e.g. 10.1007/s00216-016-0045-4) 

more comments are as follows;

Abstract section

..Although liquid chromatography-tandem mass spectrometry (LC-MS/MS) is increasingly 19 utilized across different fields, this technique is currently not used for ADA against infliximab meas- 20 urements...

There are more than 100 papers for this topic  What this contribute is ambiguous.

Page 4 line 148-175  Refs needed

Page 5 line 202-204

It is descriptive and more explanation is needed 

Page 6 line  228-246

The authors should put more data includin the relevant TICs.

The obtained results should be compared with the existing literature regarding infliximab measurements. 

Reviewer 2 Report

Authors reported a method validation study for the detection and measurement of anti-drug antibodies (ADA) against infliximab. While the conventional method is immunoassays such as RIA, authors developed and optimized a method using liquid chromatography-tandem mass spectrometry (LC-MS/MS). They incorporated the use of protein A magnetic beads to capture IgG followed by adding SIL IFX F(ab')2 for specifically binding to the IFX-ADA which were then measured indirectly. 

The method is sound. The finding is solid, and the conclusion is supported. There are a few minor comments which the authors should nonetheless address before the manuscript is accepted for publication. These are listed below.

1. Methods, 2.8 Validation 

Line 202-203: Please briefly describe what constitutes this validation method adapted by this study? 

2. Methods, 2.10 Stat analysis

Line 210-212: There were remnant plasma of patients treated with IFX at the UMCU used in this study, for cross-validation purpose. Should the heading of this subsection be revised to reflect the nature of this part of work (instead of "statistical analysis")? Also, the use of real human samples here should be accompanied with ethical approval and consensus from the patients involved. I am not sure if this information has been included in this manuscript or submission as it does not appear in the text of this version. Please consider including this information. 

3. Figure 1 caption (line 179) stated the blocking of rheumatoid factors with Kiovig® F(ab’)2. For the interest of readers from various fields, please include a brief explanation for the use of this either in the methodology or discussion section. 

4. Results: 

Line 256: "(ULOQ) of 880 arbitrary units (AU)/mL which equites to 8.8 mg/L for the RIA". -- Please explain how was this done (for the approximation)? 

5. Discussion: Although authors deployed RIA to validate the LCMS method, will there be variation from other assays such as EIA? In addition, aspecific binding from ADA against other drugs, will this be a concern or variable that may affect the validity of the test method developed? These may be meant for discussion on future work or limitation. 

Overall this is a well designed and executed study. 

Reviewer 3 Report

Dear Authors,

I suggest, this paper can be accepted in presented form. 

Warm regards

Author Response

Dear reviewer,

We would like to express our sincere appreciation for the time and effort reviewer 3 took to evaluate our manuscript.

Warm regards

Round 2

Reviewer 1 Report

The revised manuscript is now suitable for publication